# Increased Heme Oxygenase 1 Expression upon a Primary Exposure to the Respiratory Syncytial Virus and a Secondary *Mycobacterium bovis* Infection

**DOI:** 10.3390/antiox11081453

**Published:** 2022-07-26

**Authors:** Gisela Canedo-Marroquín, Jorge A. Soto, Catalina A. Andrade, Susan M. Bueno, Alexis M. Kalergis

**Affiliations:** 1Millennium Institute of Immunology and Immunotherapy, Departamento de Genética Molecular y Microbiología, Facultad de Ciencias Biológicas, Pontificia Universidad Católica de Chile, Santiago 8331150, Chile; gecanedo@uc.cl (G.C.-M.); jasoto6@uc.cl (J.A.S.); cnandrade@uc.cl (C.A.A.); sbueno@bio.puc.cl (S.M.B.); 2Centre for Biomedical Research and Innovation, Faculty of Dentistry, Universidad de los Andes, Santiago 7620157, Chile; 3Millennium Institute on Immunology and Immunotherapy, Departamento de Ciencias Biológicas, Facultad de Ciencias de la Vida, Universidad Andrés Bello, Santiago 8370146, Chile; 4Departamento de Endocrinología, Facultad de Medicina, Pontificia Universidad Católica de Chile, Santiago 8330023, Chile

**Keywords:** RSV, mycobacteria, HO-1, tuberculosis, susceptibility

## Abstract

The human respiratory syncytial virus (hRSV) is the leading cause of severe lower respiratory tract infections in infants. Because recurrent epidemics based on reinfection occur in children and adults, hRSV has gained interest as a potential primary pathogen favoring secondary opportunistic infections. Several infection models have shown different mechanisms by which hRSV promotes immunopathology to prevent the development of adaptive protective immunity. However, little is known about the long-lasting effects of viral infection on pulmonary immune surveillance mechanisms. As a first approach, here we evaluated whether a primary infection by hRSV, once resolved, dampens the host immune response to a secondary infection with an attenuated strain of *Mycobacterium bovis* (*M. Bovis*) strain referred as to Bacillus Calmette-Guerin (BCG). We analyzed leukocyte dynamics and immunomodulatory molecules in the lungs after eleven- and twenty-one-days post-infection with *Mycobacterium*, using previous hRSV infected mice, by flow cytometry and the expression of critical genes involved in the immune response by real-time quantitative reverse transcription polymerase chain reaction (RT-qPCR). Among the latter, we analyzed the expression of Heme Oxygenase (HO)-1 in an immunization scheme in mice. Our data suggest that a pre-infection with hRSV has a conditioning effect promoting lung pathology during a subsequent mycobacterial challenge, characterized by increased infiltration of innate immune cells, including interstitial and alveolar macrophages. Our data also suggest that hRSV impairs pulmonary immune responses, promoting secondary mycobacterial colonization and lung survival, which could be associated with an increase in the expression of HO-1. Additionally, BCG is a commonly used vaccine that can be used as a platform for the generation of new recombinant vaccines, such as a recombinant BCG strain expressing the nucleoprotein of hRSV (rBCG-N-hRSV). Therefore, we evaluated if the immunization with rBCG-N-hRSV could modulate the expression of HO-1. We found a differential expression pattern for HO-1, where a higher induction of HO-1 was detected on epithelial cells compared to dendritic cells during late infection times. This is the first study to demonstrate that infection with hRSV produces damage in the lung epithelium, promoting subsequent mycobacterial colonization, characterized by an increase in the neutrophils and alveolar macrophages recruitment. Moreover, we determined that immunization with rBCG-N-hRSV modulates differentially the expression of HO-1 on immune and epithelial cells, which could be involved in the repair of pulmonary tissue.

## 1. Introduction

Human respiratory syncytial virus (hRSV) is the leading cause of acute lower respiratory tract infection (ALRTI) in children under five-year-old, displaying high rates of hospitalization and over 90,000 death-related cases [1,2]. Children under two years of age, people with comorbidities, such as cardiac and pulmonary affections, and the elderly, are the most susceptible to developing hRSV bronchiolitis and pneumonia [3]. HRSV infections cause airway inflammatory hyperresponsiveness characterized by the secretion of pro-inflammatory cytokines, including interleukin (IL)-6, IL-4, IL-13, and tumor necrosis factor (TNF)-α, leading to the recruitment of innate leukocytes, such as neutrophils, as well as adaptive immune cells including T and B cells [4].

The consequences associated with hRSV infection during the early stages of infection include poor adaptive immunity that allows reinfections [5,6] and increased susceptibility to the development of allergies [7], post-bronchiolitis wheeze (PBW), and asthma [8]. Another important consequence of hRSV infections is the frequent establishment of secondary bacterial infections, such as *Streptococcus pneumoniae* (*S. pneumoniae*), which can lead to more severe clinical symptoms of pulmonary pathology, and are considered an essential factor that contributes to mortality rate [9,10,11,12]. Different studies have suggested that primary respiratory viral infections can induce host susceptibility to secondary infections, either acute or concomitant bacterial infection (*S. pneumoniae*) or established bacterial infection after viral clearance, such as *Mycobacterium tuberculosis* (*M. tuberculosis*) [13]. However, the susceptibility that an initial infection with hRSV promotes for secondary infections with *M. tuberculosis* has not been evaluated. Here we sought to evaluate whether hRSV infection shapes the lung immune response and promotes pulmonary inflammatory hyperresponsiveness in a chronic mycobacterial infection model.

The *Mycobacterium bovis* (*M. Bovis*) strain, referred to as Bacillus Calmette-Guerin (BCG), is an attenuated version of a known species belonging to the *M. tuberculosis* complex [14]. Although infections with BCG do not recapitulate all pathological features observed with *M. tuberculosis*, BCG is a valuable model for studying anti-mycobacterial immune responses mimicking important aspects of bacilli-host interaction [15,16]. Moreover, BCG provides a functional model for chronic infections as viable bacilli can persist for up to 10 months in the lungs [17].

Heme Oxygenase (HO)-1 is an enzyme that catalyzes a reaction in which the heme group turns into carbon monoxide (CO), biliverdin, and free iron [18]. HO-1 promotes heme group homeostasis and has a cryoprotective role against tissue damage. Additionally, it has been reported that the activity of HO-1 has an antiviral effect on pathogens such as hRSV, among others [19,20,21]. In the case of mycobacteria, the function of HO-1 in the host is induced by the dormancy mechanism activated by the mycobacteria, promoting an anti-inflammatory response that avoids the immune system and ensures their survival [22,23]. The expression of HO-1 is controlled by a transcription factor named nuclear factor erythroid 2-related factor (Nrf2), which can be activated as a consequence of stimuli such as hypoxia [24]. The OX-2 glycoprotein membrane (CD200) is an important molecule expressed on the surface of alveolar epithelial cell type II, which can bind to the receptor CD200R on the surface of alveolar macrophages [25]. The union of CD200 with its receptor on the alveolar macrophages leads to their inhibition, downregulating the inflammatory response in the pulmonary epithelium [25]. By doing this, CD200 can regulate the airway immunological response against infections.

As mentioned above, hRSV-infection can cause airway inflammation and hyperresponsiveness [4]. However, the link between hyperresponsiveness of the airways and long-term pulmonary sequels regarding *Mycobacterium* infections is unclear. Therefore, we evaluated whether the lungs of hRSV-infected mice are immunologically susceptible to secondary bacterial colonization along with pulmonary pathology. With this aim, we performed long-term bacterial colonization using BCG as the *Mycobacterium* infectious model after a primary infection with hRSV in C57BL/6 mice [26,27]. Additionally, HO-1 induction during hRSV-infection produces an anti-inflammatory environment and reduction of viral loads [20]. Since the HO-1 effect in BCG immunization has not been evaluated, we decided to test whether the effect previously described was increased using a BCG-based vaccine prototype against hRSV. This study aimed to determine the contribution of HO-1 during a mycobacterial infection that followed an infection by hRSV and after an immunization scheme followed by an hRSV challenge.

## 2. Materials and Methods 

### 2.1. Ethics Statements

All experimental protocols followed guidelines from the Sanitary Code of Terrestrial Animals of the World Organization for Animal Health (OIE, 24. Edition, 2015) and were reviewed and approved by the Scientific Ethical Committee for Animal and Environment Care of the Pontificia Universidad Católica de Chile (Protocol number 160915010 and 160405005). All mouse experiments were conducted in agreement with international ethical standards and according to the local animal protection law number 20,800.

### 2.2. Viral Propagation and Titration

Human Epidermoid carcinoma strain 2 (Hep-2) cell line (American Type Culture Collection, CCL-23^TM^) (American Type Culture Collection, CCL-7^TM^) was used to propagate hRSV serogroup A2, strain 13018–8, a clinical isolate from Instituto de Salud Pública de Chile [28]. Hep-2 monolayers were grown in T75 flasks with Dulbecco’s Modified Eagle Medium (DMEM) (Life Technologies Invitrogen, Carlsbad, CA, USA) supplemented with 10% fetal bovine serum (FBS) (Gibco Invitrogen Corp, Carlsbad, CA, USA) until 80–90% confluency. Flasks containing 5 mL of DMEM 1% FBS for infection with hRSV, the viral inoculum with 2 × 10^5^ plaque formation units (PFU), were incubated at 37 °C. After 2 h of virus adsorption, supernatants were replaced with fresh DMEM 1% FBS medium and incubated for 48 h (i.e., until cytopathic effects were detectable). For harvesting, cells were scraped, and the flask content was pooled and centrifuged first at 300× *g* for 10 min and then at 500× *g* for 10 min to remove cell debris. Using the same harvesting protocol, supernatants of non-infected cells were collected and used as the non-infectious control (referred to from here on as Mock). Viral titers of supernatants were determined by immunocytochemistry in 96-well plates with Hep-2 cells, as described previously [29,30].

### 2.3. Mycobacterium Bovis-BCG-Culture, and Storage

The BCG Danish 1331 strain was grown in medium 7H9 (Sigma-Aldrich, Saint Louis, MO, USA, M0178-500G), a specific mycobacteria broth [30], supplemented with 10% Middlebrook oleic acid, albumin, dextrose, and catalase (OADC) Growth Supplement (Sigma-Aldrich, M0678-1VL), with constant stirring at 120 rpm until reaching an OD600 nm equal to 0.8. At this point, the mycobacteria culture was washed three times with 1X PBS-0.05% Tween 80, resuspended with 1X PBS-glycerol 50% at a final concentration of 1 × 10^6^ colony-forming units (CFU)) per vial and frozen at −80 °C until their use. For the infection, BCG vials were centrifuged at 14,000× *g* and resuspended in PBS for intranasal administration. 

### 2.4. Mouse Immunization and Viral Infection

The effect of a recombinant BCG (rBCG) strain in the modulation of HO-1 was evaluated by immunization with rBCG expressing the nucleoprotein of hRSV (rBCG-N-hRSV) as described next. Six to eight-week-old BALB/cJ mice were immunized by sub-cutaneous 1 × 10^8^ CFU of BCG WT or rBCG-N-hRSV in a final volume of 100 μL per dose at days 0 and 14. Twenty-one days after immunization, mice were intraperitoneally anesthetized and challenged by intranasal infection with ~1 × 10^7^ PFU of hRSV A2, strain 13018-8. On days 7 and 14 post-infection, mice were euthanized. rBCG production was performed as described previously [30].

### 2.5. Mouse Viral and Mycobacterial Infections

Two infection schemes were conducted to determine the consequences of hRSV infection and the effect of a subsequent infection. The short scheme was performed to evaluate if the inoculation with mycobacteria could infect and damage the pulmonary tissue a few days after the clearance of hRSV (day 10 post-infection with hRSV). Euthanasia was performed 11 days after the inoculation with mycobacteria since the alveolar tissue repair was not complete on this day. For this reason, it would be expected to find an effect induced by the administration of mycobacteria. The choice of this day was mainly due to the slow replicative cycle of this bacterium, which makes it difficult to carry out these tests at earlier times. The long scheme was performed to evaluate if, a significant number of days after the clearance of hRSV (day 21 post-infection with hRSV), the inoculation with mycobacteria could infect and damage the pulmonary tissue. During the day of the inoculation, the cell target of mycobacteria alveolar macrophages was replaced, allowing the mycobacteria more time to proliferate and cause an effect on the lung. Euthanasia was performed 21 days after the inoculation with mycobacteria since, on this day, the alveolar tissue repair is not complete. Six to eight-week-old C57BL/6J mice received an intranasal infection with 1 × 10^7^ PFU of hRSV A2 strain 13018-8 of 100 μL per mouse. After 10- or 21-days post-infection (dpi), mice were intranasally instilled with 1 × 10^6^ CFU/mice of BCG. After 11- and 21-days post inoculation with BCG, mice were euthanized for collection of lung samples, bronchoalveolar lavage (BAL), and mediastinal lymph nodes. The controls were mock and vehicle (Sauton diluent) for the inoculation with hRSV and BCG, respectively. The administration of the first and second inoculation and their description are described in Table 1.

### 2.6. Evaluation of hRSV-Associated Disease Parameters

To determine the infiltration of polymorphonuclear cells, BAL was collected as previously described [29] and stained with anti–CD11b PerCP-Cy5.5 (clone M1/70, BD Pharmingen, San José, CA, USA), anti-CD11c APC (clone HL3, BD Pharmingen), anti-IA/IE APCcy7 (clone M5/114.15.2, Biolengend), anti-Singlec F PE CF594 (clone E50-2240, BD Bioscience, San José, CA, USA), anti-Ly6C BV605 (clone HK1.4, Biolegend, San José, CA, USA) and anti-Ly6G FITC (Clone 1A8, BD Pharmingen) antibodies. As previously described, viral loads were detected in the lungs by real-time quantitative reverse transcription polymerase chain reaction (RT-qPCR) [28,30,31]. In addition, lung samples were stored in 4% paraformaldehyde solution (PFA), maintained at 4 °C, embedded in paraffin, cut, and stained with H&E as previously described [29]. The BCG count after 21 dpi was performed by seeding 1000 μL of lung homogenate in 7H10 plates and incubation for 14 to 21 days at 37 °C with 5% CO2. To evaluate the presence of HO-I in the lungs by flow cytometry, the lungs were incubated with collagenase IV for 30 min at 37 °C with agitation (120 rpm). Then the cells were homogenized using a 70 µm cell strainer. The cells were incubated with ammonium-chloride-potassium (ACK) lysis buffer for 5 min and centrifugated 300 g for 5 min at 4 °C, and then stained with α-CD45-BV510 (clone 30-FL1, BD Horizon, San José, CA, USA), α-CD11c APC (clone HL3, BD Pharmingen), α-IA/IE-V500 (clone M5/114.15.2, BD Pharmigen), α- CD326(Ep-CAM) PE (clone G8.8, Biolegend). Then, for HO-1 intracellular staining, fixed cells were incubated with anti-mouse HO-1 monoclonal antibody (mAb) (Abcam, UK) in permeabilization buffer (1% saponin, 10% FBS in PBS) for 45 min at 4 °C. For all antibody dilutions, 1 µL of antibody was diluted in 500 µL of a buffer PEB (1X Phosphate Buffered Saline (PBS)-0.5% Bovine Serum Albumin (BSA)-0.2mM Ethylenediaminetetraacetic acid (EDTA)). Data were acquired in an LSRFortessa X2-0 cytometer (BD Biosciences) and analyzed using FlowJo v10.0.7 software (BD Biosciences). The gating strategy for detecting immune cells is shown in Appendix A.

### 2.7. Lung Histopathology Analyses

Before proceeding with BAL collection, the major bronchus of the left lung was clamped using 10 cm Kelly hemostatic forceps to perform histopathology analyses without significantly altering tissue architecture. After obtaining the BAL from the right lung, the left lung was fixed with 4% paraformaldehyde, then paraffin-embedded using a Leica ASP300S enclosed, automatic tissue processor (Leica Microsystems, Wetzlar, Germany). Then, 5 µm-thick tissue sections were obtained using a Microm HM 325 Rotary Microtome (Thermo Scientific, Waltham, MA, USA) before being mounted and stained for histopathology analyses using H&E stain. A histopathological score was used to measure structural alterations in lung sections of control and infected animals [32,33]. The histopathological score was as follows: 0 = normal tissue morphology, normal alveolar architecture, connective tissue associated with bronchi (slight presence of immune cells); 1 = alveolar spaces are reduced, and there are immune cells present; 2 = bronchoalveolar involvement is defined as reduced alveolar spaces and high infiltration of immune cells (neutrophils and lymphocytes) within and surrounding the airways, including bronchi; 3 = pulmonary consolidation is evidenced by loss of alveolar spaces, bronchial walls thickening or bronchial collapse, and high cellular infiltration. In addition, Ziehl-Neelsen (ZN) staining was performed according to standard protocols [34,35]. In brief, the major bronchus of the left lung section was dewaxed by washing with decreasing alcohol concentrations (from 96% to 70% ethanol), heat-fixed, then stained with carbol-fuchsin (Bacto TB Carbolfuchsin KF, Becton-Dickinson, Sparks, MD, USA) for 4 min, washed, and incubated with 3% hydrochloric acid (HCl) until the stain was completely dissolved. Counterstaining was performed with brilliant green (Bacto TB Brilliant Green K, Becton- Dickinson, Sparks, MD, USA) for 20 s. Sections were air-dried after thorough washing. 

### 2.8. Relative Expression by RT-qPCR

Quantitative real-time RT-qPCR total RNA was isolated from lung tissues collected using the Trizol reagent according to the manufacturer’s instructions (Thermo Fisher Scientific). Complementary DNA (cDNA) synthesis from total RNAs was performed using the iScript^TM^ Reverse Transcription Supermix for RT-qPCR (Bio-Rad, Hercules, CA, USA) and random primers. RT-qPCR reactions were carried out using a StepOne plus thermocycler (Applied Biosystems, Waltham, MA, USA). The abundance of *ho-1* and *nrf2* mRNAs were determined by relative expression to the respective housekeeping gene (*β-actin* gene) by the 2-ΔΔ threshold cycle (ΔΔCt) method [36]. The RT-qPCR assays performed had 100% efficacy. For *n-hRSV* gene expression, absolute quantification data were expressed as the number of hRSV N-gene copies for every 5 × 10^3^ copies of the *β-actin* transcript, as previously described [28,37]. The choice of using the *β-actin* gene for a single reference gene is based on present high stability [31,38]. The primers used can be found in Table 2.

### 2.9. Statistical Analyses

All statistical analyses were performed using GraphPad Prism version 6.0 Software. Statistical significance values and analyses are detailed in each figure legend. For Figure 1, only a normal distribution was observed for the data in Figure 1B. The data from Figure 1C–F showed a non-normal distribution, so non-parametric Mann-Whitney tests were performed. One-way ANOVA tests with a post hoc Tukey test were performed in Figure 2, Figure 3, Figure 4 and Figure 5 and Appendix A since the data showed a normal distribution. Two-way Anova with a post hoc Dunnett’s multiple comparisons test were performed to compare the kinetics of the weight curves of Figure 2A and Appendix A. Figure 1A used a two-way ANOVA with a post hoc Šídák’s multiple comparisons test.

## 3. Results

### 3.1. HRSV Infection Induces de Expression of Immunomodulatory Molecules

Our study model evaluated different disease and cellular infiltration parameters to characterize the inflammatory immune response induced by hRSV. A significant reduction in viral loads was observed during previous studies using a mouse infection model by day 7 post hRSV instillation [39]. Therefore, we measured weight loss and viral load by quantitative RT-PCR from the lungs of mice at 10 dpi and used as positive controls (C+) the viral load at 3 dpi (Figure 1A,B). No significant differences were found in the weight loss after the infection compared to mock-treated mice (Figure 1A). As shown in Figure 1B, animals showed levels of *n-hRSV* RNA close to non-detectable, indicating an adequate viral clearance by day 10 post-challenge, making it an appropriate inoculation time point for subsequent BCG challenge. As expected, only a significant difference was found in the positive control with respect to the groups evaluated (*p* = 0.0001). These results correlated with a low percentage of neutrophils recruited in the lungs (Figure 1C). Additionally, to evaluate whether the resolution of the infection was affected by anti-inflammatory modulators, the expression of *ho-1*, nrf2 related to oxidative stress, and *cd200* genes were measured by RT-qPCR. At 10 dpi, we observed a significant increase in *ho-1* (*p*= 0.0152) and *cd200* (*p* = 0.0244) relative expression in convalescent mice lungs compared to control animals. However, no differences were observed in the transcription factor controlling *ho-1* gene expression *nrf2* between the groups (Figure 1F). We have recently shown that HO-1 upregulation has an antiviral protective effect in the lungs by limiting viral replication in the tissue and epithelial cells, as well as by modulating the immunogenicity of antigen-presenting cells (APC), such as dendritic cells (DC) and alveolar macrophages [20,40]. Given these data, it could be suggested that upon viral clearance, the high levels of *ho-1* expression in the lower respiratory tract might play a role in modulating the immunity to secondary infections.

### 3.2. Previous hRSV Infection Causes Further Long-Term Susceptibility to Mycobacterium Bovis-Driven Pneumonia

To further evaluate whether animals that have cleared hRSV from the lungs remain immunocompetent to clear secondary infections, control (mock-instilled) and hRSV-convalescent animals were challenged with *M. bovis* BCG after 10 dpi. BCG inoculation was given intranasally to mice using a saline solution as a control (Figure 2A). As shown in Figure 2B, hRSV-convalescent mice were inoculated with BCG (*hRSV-BCG*), but not their relevant controls (*Mock-BCG*), showed increased weight loss at day 13 post-infection. Statistical differences were found along the kinetics for *hRSV-BCG* vs. *Mock-Vehicle* or *Mock-BCG* groups (*p* ≤ 0.0001). This weight loss can be associated with the infection with BCG rather than hRSV persistence, as evidenced by the absence of detectable viral loads at day 21 post-infection (Figure 2C). As expected, only a significant difference was found in the positive control with respect to the groups evaluated (*p* = 0.0002). Additionally, this was associated with more severe histopathological scores in both groups infected with BCG. Careful histopathological scoring did not show any significant difference in the semiquantitative blinded scoring of both BCG-infected groups despite a more marked thickening of alveolar walls (Figure 2D,E). The more severe lung pathology observed at 21 dpi in the *hRSV-BCG* group was characterized by the thickening of the lung parenchyma due to interstitial inflammation in discrete and well-defined foci developed around bronchi and alveolar sacs (Figure 2E), which showed a significant difference compared to *Mock-Vehicle* (*p* = 0.00219). Although less severe, a mild inflammation of the lung parenchyma was observed with foci in the bronchi in mice pre-treated with Mock and then inoculated with BCG (*Mock-BCG*) (Figure 2D,E). Additionally, the maintenance of the pulmonary architecture and no significant lung inflammation were observed in both vehicle-inoculated controls (*Mock-Vehicle* and *h**RSV-Vehicle* groups). Mycobacteria CFUs were detected in the lungs of animals previously inoculated with either mock or hRSV (Figure 2F). Consistent with the histopathological score, only the *hRSV-BCG* group showed a significant increase of BAL neutrophils (Ly-6C^−^ CD11b^+^ Ly-6G^hi^) compared to the *Mock-Vehicle* group (*p* = 0.0445) (Figure 2G). Accompanying neutrophils, we observed no significant differences in BAL monocytes (Ly-6C^+^ CD11b^+^ Ly-6G^−^) and eosinophils (CD11b^+^ Siglec F^+^) in all groups of mice (Figure 2H,I, respectively). Next, we analyzed other lung immune cell populations by flow cytometry, such as interstitial and alveolar macrophages. Consistent with the infiltration of neutrophils in BAL, alveolar macrophages, defined as CD11b^−^ CD11c^+^ Siglec-F^+^ [41], showed a significant increase only in *hRSV-BCG* compared to both *Mock-Vehicle* (*p* = 0.0062) and *Mock-BCG* (*p* = 0.0159) groups (Figure 2J). A slight, non-significant increase was observed in the *hRSV-Vehicle* group (Figure 2J). No significant increase in interstitial macrophages (CD11b^+^CD11c^+^Siglec-F^−^) was detected between groups, but a slight increase was found in the *hRSV-BCG* group. (Figure 2K). 

To assess whether the increased susceptibility to *M. bovis* BCG induced by a previous hRSV infection is short-lived, we evaluated a long scheme in which the second inoculation and the euthanasia were performed 21 and 42 days after the first inoculation, respectively (Table 1). Interestingly, the results obtained from these trials showed that there were no significant changes between those reflected in the parameters of disease or inflammation evaluated in the different groups of animals (Appendix A), except for more significant damage related to the histological score (Appendix A).

Following our detection of viable bacilli in the lungs of BCG-infected animals, we performed ZN staining of the lung sections shown in Figure 3A to assess whether the significant increase in pathology and infiltration of neutrophils correlated with the increased presence of dormant bacilli, which might not be detected by ex vivo culture of lung homogenates. In agreement with the low numbers of CFU detected in BCG-infected animals, no active *M. bovis* was found in the lungs of both BCG-infected groups of mice (Figure 3A). However, a population of macrophages that reacted with the stain was observed only in the lungs of *hRSV-BCG* treated mice but not in the other treatments (red arrows). Since ZN staining binds to mycolic acids, the observation of weak ZN staining suggests the existence of either myeloid cells processing remainder mycobacterial components, or macrophages containing bacilli with a low metabolic rate (i.e., under dormancy) [42].

Since HO-1 induction has been associated with both impairment of antigen processing by APC [40] and bacilli dormancy [23], we sought to evaluate whether the detection of ZN-stained cells was associated with the modulation of HO-1 expression in the lungs. A significant increase of almost 50% in the relative expression of HO-1 was observed only in *h**RSV-BCG*, as compared to *mock-vehicle* (*p* = 0.0357), *hRSV-vehicle* (*p* = 0.0263) or *mock-BCG* (*p* = 0.0239) control groups (Figure 3B). No significant differences were observed in the expression of Nrf2, suggesting that HO-1 up-regulation might imply other transcription factors [43] (Figure 3C). 

No significant changes were observed in the determination of dominant BCG in the lungs of the long scheme animals, nor were differences in the expression levels of HO-I or NRF2 (Appendix A).

### 3.3. Characterization of the Inflammatory Landscape of the Lungs

As mentioned above, CD200 can induce an inhibitory signal in alveolar macrophages, among other APC, downregulating the inflammation in the pulmonary tissue [25]. No significant differences in the expression of the epithelial anti-inflammatory molecule CD200 were observed between the groups of the short scheme (Figure 4A), suggesting that the absence of CD200 upregulation might support the observed lung pathology [43]. Additionally, the relative expression of pro-inflammatory cytokines IFN-γ, IFN-β, and IL-6 was evaluated, but no significant differences were found for any of these molecules (Figure 4B–D). Only a slight increase in the relative expression of IFN-γ was detected in the *hRSV-BCG* compared to the other groups (Figure 4B). For the long scheme, an increase in the relative expression was observed for all molecules evaluated. Still, only significant differences were found for the expression of *ifn-γ* for the *hRSV-BCG* group compared to *mock-vehicle* (*p* = 0.0094) (Appendix A).

### 3.4. rBCG-N-hRSV Immunization Promotes a Higher Induction of HO-1 on Epithelial Cells than in DCs at Late Infection Times

BCG is commonly used as a vaccine to prevent infections by *M. tuberculosis* and is often used as a vector for generating new recombinant vaccines [14]. Therefore, we decided to evaluate if the immunization with an rBCG-N-hRSV can modulate the expression of HO-1 in hRSV-infected mice. After 7 and 14-dpi with hRSV, mice were euthanized, and both epithelial and DCs from the lungs were stained to evaluate the presence of HO-1 (Figure 5). The HO-1 expression observed on day 7 post-infection in DCs showed a non-significant decrease in the *rBCG-N-hRSV* immunized mice compared to mock-treated mice. A similar result was observed in the rBCG-N-hRSV group in *BCG-WT* and hRSV-infected mice (Figure 5B). In contrast, when HO-1 was evaluated in epithelial cells, the mock-treated group showed the lowest non-significant levels of expression as compared to the rest of the infected groups (*mock-treated* compared to *hRSV-infected* (*p* = 0.9680), *BCG-WT* (*p* = 0.2232), *rBCG-N-hRSV* (*p* = 0.733)) (Figure 5C). When the HO-1 expression was evaluated on day 14 post-infection in DCs, similar results were obtained compared to day 7 post-infection. Interestingly, although the behavior of the groups was similar to that previously reported, the magnitude order was 4 to 5-fold less (Figure 5D). On the other hand, epithelial cells showed a different effect in the HO-1 expression on day 14 post-infection, demonstrating a different behavior distinct from the HO-1 expression in epithelial cells on day 7 post-infection. In this line, the HO-1 expression was higher in both *mock-treated,* and *rBCG-N-hRSV* immunized mice than in hRSV-infected or *BCG-WT* groups (Figure 5E), suggesting that the immunization with rBCG-N-hRSV promotes a lasting induction of HO-1 through time.

## 4. Discussion

The infection with hRSV is the principal cause of ALRTI, and it has been demonstrated that it can predispose to secondary lung infections [1]. The effects of primary viral infections on the outcome of concomitant secondary acute bacterial infections have only been studied in mice [13], such as *S. pneumoniae,* which induces a more severe pulmonary pathology [11,44]. Some studies have demonstrated that the immunopathology generated during coinfections of viruses, with either other viruses or bacteria, depends on the primary viral infection stage [14,42]. Therefore, it is important to establish the stage of the viral infection that will be evaluated. It has been reported that the administration of *S. pneumoniae* following the peak of hRSV replication in the lungs increases bacterial burden (analyzed as CFU and infiltration of inflammatory cells [13]. However, the link between hyperresponsiveness of the airways and long-term pulmonary sequels is not clear, although two possible explanations have been suggested. First, it was proposed that hRSV produces long-term damage in the airway epithelium, which may favor the infection by other pathogens [13,45]. In this line, studies with other respiratory viruses, such as influenza A virus, have shown a detrimental effect of the viral infection on the survival rate and clearance of *M. tuberculosis* in mice after viral clearance was demonstrated in the lungs [46]. As a second hypothesis, it was proposed that hRSV-infection exerts an immunomodulatory effect on the lungs, which predisposes to allergy and asthma [47]. To the best of our knowledge, no studies address whether hRSV infection exerts long-lasting effects on pulmonary immune responses, especially after the resolution of the initial viral insult.

As mentioned previously, the infection with BCG is a valuable model for studying anti-mycobacterial immunity, including forming granulomatous lung lesions and acquiring a paucibacillary state [15,16]. The mouse model of intranasal BCG infection is suitable for addressing whether hRSV blunts the desired type 1 T helper (Th1)-driven anti-mycobacterial immune responses, which correlate with protection to other mycobacterial species [48,49]. Furthermore, it has been described that hRSV-effects can last up to 42 dpi in the BALB/c mouse model [50] and up to 20 dpi in C57BL/6 mice [51].

This work aimed to unravel whether a previous infection with hRSV can elicit an immune response that facilitates the mycobacterial infection in a murine model. Therefore, we used an intranasal administration of attenuated *M. bovis* strain, BCG, to establish our subsequent infection model. Since BCG has been used as an infection model of TB [49,52,53], here we used an intranasal administration of BCG after the clearance of the infection with hRSV in mice to elucidate if this viral infection promotes an immune response that favors a secondary BCG infection. One of the significant discoveries in this work was the detection of brown structures in the lung epithelium in the *h**RSV-BCG* group, where it was possible to identify the presence of mycobacteria in the acid-fast staining (Figure 3A). One hypothesis is that these brown spots, present only in the *h**RSV-BCG* group, are lipid droplets that lose the acid-fast staining, as the color is brown and even considered orange or red [42] (Figure 3A and Appendix A). These drops of lipids can result from a mechanism of *Mycobacterium* when it remains dormant, considering that the *Mycobacterium* is surrounded by a monolayer of phospholipids and uses these structures as a primary source of carbon [42]. This stage occurs when the bacteria are in a context that promotes hypoxia, such as when the pulmonary tissue is not yet fully recovered during respiratory infections, such as hRSV [54,55]. The absence of complete recovery in the pulmonary tissue can be suggested due to the significant increase of CD200 (Figure 1F) and HO-1 relative expression (Figure 1E), which promote an anti-inflammatory environment [55]. Additionally, we can suggest that both CD200 and HO-1 might play a role in restructuring the lung’s epithelial environment after 10 dpi with hRSV [55]. After inoculating the mice with BCG, further damage was observed, and the cell infiltration remained increased, as shown in Figure 2. Similar results are observed in Appendix A. The mycobacteria acquire a dormant state that increases the expression of HO-1 to produce carbon monoxide and promote an anti-inflammatory state in the host cells [23,56]. Accordingly, the increased detection of HO-1 and the possibility of macrophages containing bacilli in a dormant state (Figure 3) can be related to the presence of mycobacteria in the epithelium [57,58]. In this context, tissue oxygenation is impeded, favoring the conditions in which the *Mycobacterium* is put into a dormant state through triglyceride metabolism [59,60,61]. Unlike hRSV, which produces ALRTI, BCG infection produces a less diffuse pulmonary pathology and, therefore, a less pronounced body weight loss (Figure 2B,E). 

One of the limitations of this study was that the observation of mycobacterial infection was performed over an unprolonged period. With an even longer-term scheme, it would be possible to observe how the mycobacteria behave at cellular levels and if the cellular populations change on this day. This is because the airway epithelium has a cell replacement every twenty days but not in the alveolar macrophages, which can stay in the lung for years [62,63,64]. However, it must be considered that the alveolar macrophages will remain in the lungs for long periods if they are not destroyed by pathogenic infections [65]. Even more, some respiratory viruses, such as influenza virus, can promote changes in the alveolar macrophages to induce an extended antibacterial response [66]. In addition, more days allow for identifying the mycobacteria by acid-fast staining, which is correlated with the CFUs counts (Figure 2F and Figure 3A). This study also used BCG mycobacteria as a model, making it possible to use it in a murine model and address scientific questions with an attenuated pathogen. However, this work proves that it serves as a model for research on secondary *Mycobacterium* infections despite not having the same effect and clinical symptoms as an infection with *M. tuberculosis*. 

In addition, relative cytokine expression was evaluated for *ifn-γ*, *ifn-**β*, and *il-6* (Figure 4B–D), which participate in the control of mycobacterial infections and other bacterial infections [67,68]. Thus, *ifn-**γ, ifn-**β,* and *il-6* are related to mycobacteria colonization in these schemes. In the case of *ifn-**γ* in the short scheme of infection, the relative expression was more elevated in the *hRSV-BCG* group than in *Mock-BCG* and the other groups (Figure 4B). Interestingly, in the short infection scheme, the hRSV-BCG group had a lower relative expression of *ifn-β* than the *hRSV-Vehicle* (Figure 4C). Still, in the long infection scheme, only the *hRSV-BCG* group had a higher relative expression of this gene than the *hRSV-Vehicle* (Appendix A). *ifn-**β* is a molecule that helps delay the beginning of mycobacterial infection [67], and as such, the data observed on the long scheme might be explained due to viral clearance. However, in the long scheme, we can see a similar phenomenon to *ifn-**γ*, where the group with the highest expression was in the groups with mycobacteria inoculation (Appendix A), which could be attributed to the presence of mycobacteria. This increase in the expression of *ifn-γ* could be associated with the presence of alveolar macrophages, which are sentinels in the pulmonary epithelium that promote a favorable environment [69,70]. Both the short (Figure 4D) and long (Appendix A) schemes presented similar relative expressions of *ifn-**γ*. Lastly, the increased relative expression of *il-6* does not inhibit the growth of mycobacteria, even though it has been reported that an increased secretion helps to protect the host against mycobacteria [68]. These data would imply that the presence of mycobacteria activates the expression of these cytokines.

We have previously shown the anti-inflammatory effect of HO-1 induction in vivo and in vitro [20]. Here we extend these observations by showing that at 7 dpi with hRSV, mice pre-vaccinated with BCG showed an elevated expression of HO-1 in DCs, but the group treated with mock had the highest HO-1 within hRSV and BCGs-groups (Figure 5B). A similar effect was found at day 14 dpi, but the activation of HO-1 was four to five-fold less than previously reported (Figure 5D), and this HO-1 in DCs elevated just in the mock group with respect to hRSV-infection was reported [20]. Interestingly, the effect of HO-1 in epithelial cells differed from that observed in DCs. Here, at 7 dpi, all hRSV-infected groups had a similar HO-1 response to the mock-treated group (Figure 5C). However, this effect was different at 14 dpi, where only rBCG-N-hRSV immunized mice had high HO-I levels, similar to the mock-treated group (Figure 5E). This last statement might suggest that hRSV-N-BCG promotes a prolonged state of HO-I activation in epithelial cells that helps eliminate the virus and recover damaged tissue [71]. No changes in the MFI of HO-I were found between the groups at different times (data not shown). Interestingly, it has been reported that the infection with hRSV can modify the methylation profile of immune cells and epithelial cells, promoting the secretion of Th2 cytokines and viral replication [72]. Since the activation of Nrf2 can be regulated epigenetically, it would be interesting to evaluate whether hRSV infection can modulate the activation of Nrf2 in an epigenetically-manner [73]. In this sense, a limitation of our article is that we did not explore whether a primary infection with hRSV can modulate epigenetic changes that may participate in the resolution of the disease against subsequent administration with *M. bovis*.

## 5. Conclusions

Pulmonary infections modulated by hRSV negatively modulate the respiratory and immune response, which promotes a sub-sequential *M. bovis* exposure, leading to inefficient mycobacterial clearance and increased host inflammatory response. The clearance is impeded by a possible dormancy state established by mycobacteria. Additionally, the mycobacteria promote HO-1 expression activating the dormancy state. In other words, hRSV promotes the development of pulmonary-tuberculosis-like in mice by increasing lung inflammation and the survival of infecting bacilli.

## Figures and Tables

**Figure 1 antioxidants-11-01453-f001:**
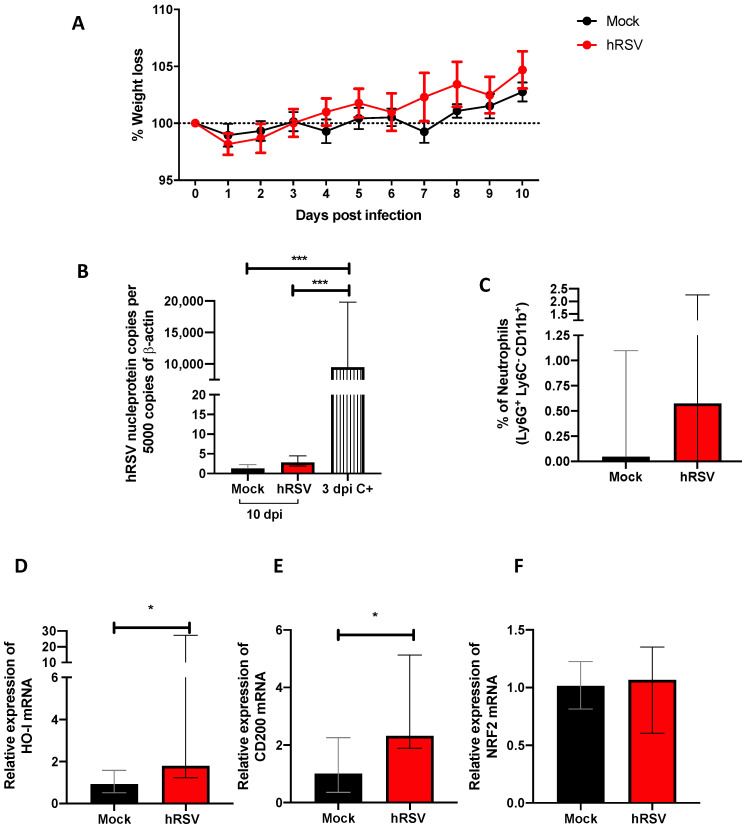
Evaluation of infection, inflammation, and immunomodulatory parameters from human respiratory syncytial virus (hRSV)-infected mice. (**A**) Body mass loss of C57BL/6 mice infected with 1 × 10^7^ plaque formation units (PFU) of hRSV A2 for ten days. All the following parameters were measured at day 10 post hRSV infection. (**B**) Determination of viral load through specific real-time quantitative reverse transcription polymerase chain reaction (RT-qPCR) for hRSV. (**C**) Neutrophils (**D**) Heme Oxygenase (HO)-1, (**E**) OX-2 glycoprotein membrane (CD200), and (**F**) nuclear factor erythroid 2-related factor (NRF2). Data are shown as median ± interquartile range of at least two independent experiments with three animals per group. (**B**) One-way ANOVA was performed with a post-hoc Tukey test. (**C**–**F**) t-student was performed with the Mann-Whitney U test (* *p* < 0.05; *** *p* ≤ 0.001). Created with BioRender.com.

**Figure 2 antioxidants-11-01453-f002:**
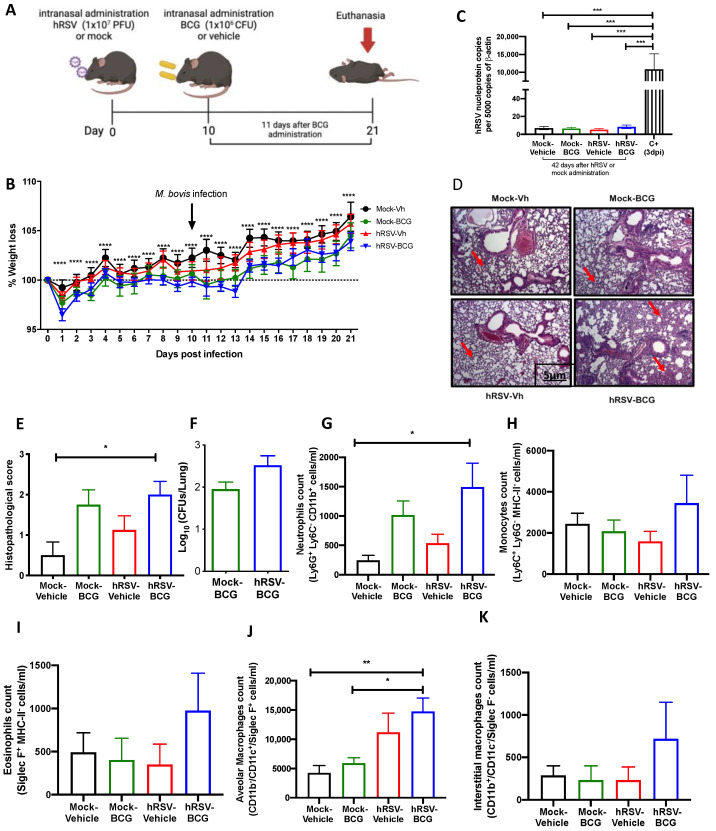
Evaluation of infection parameters from primary infection with a human respiratory syncytial virus (hRSV) and 11 days post-infection with Bacillus Calmette-Guerin (BCG). (**A**) Scheme of infection in mice. (**B**) Body mass loss of C57BL/6 mice infected with 1 × 10^7^ plaque formation units (PFU) of hRSV A2 and a subsequent challenge with BCG intranasal at day 10 post-infection with hRSV. The significant difference corresponds to a two-way ANOVA of multiple comparisons between *hRSV-BCG* versus *mock-vehicle* and hRSV-BCG versus hRSV-vehicle. (**C**) Determination of viral load through specific real-time quantitative reverse transcription polymerase chain reaction (RT-qPCR) for hRSV. (**D**) Lung tissue sections were stained with Hematoxylin and Eosin (10× magnification). (**E**) Histopathological score. (**F**) Bacterial load in the BCG-groups. (**G**–**K**) Flow cytometry analyses of bronchoalveolar lavages (BAL) from mice infected with *Mycobacterium bovis (M. bovis)*. The figure shows the absolute cell numbers for neutrophils (**G**), monocytes (**H**), eosinophils (**I**), alveolar macrophages (**J**), and interstitial macrophages (**K**) in BAL of *M. bovis*-infected mice. Data are shown as means ± SEM of three independent experiments with 3–4 animals per group. One-way ANOVA was performed with a post-hoc Tukey test. (* *p* <0.05; ** *p* < 0.01; *** *p* ≤ 0.001; **** *p* ≤ 0.0001).

**Figure 3 antioxidants-11-01453-f003:**
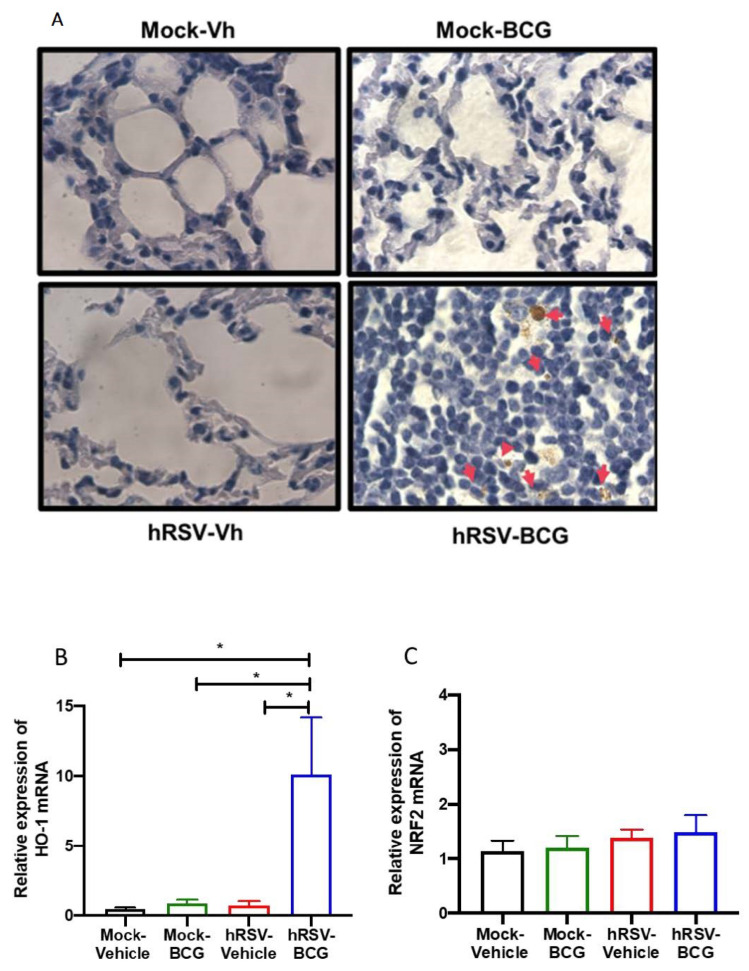
Determination of mycobacteria and Heme Oxygenase (HO)-1 activity during a short infection scheme with the human respiratory syncytial virus (hRSV) and Bacillus Calmette-Guerin (BCG). (**A**) Acid-fast staining of *Mycobacterium bovis*-infected mouse lungs collected on day 11 post-inoculation with BCG (100× magnification). Quantification by real-time quantitative reverse transcription polymerase chain reaction (RT-qPCR) of *ho-1* (**B**) and the nuclear factor E2-related factor 2 (*nrf2*) (**C**). Data are shown as means ± SEM of three independent experiments with 3–4 animals per group. One-way ANOVA was performed with a post-hoc Tukey test (* *p* < 0.05).

**Figure 4 antioxidants-11-01453-f004:**
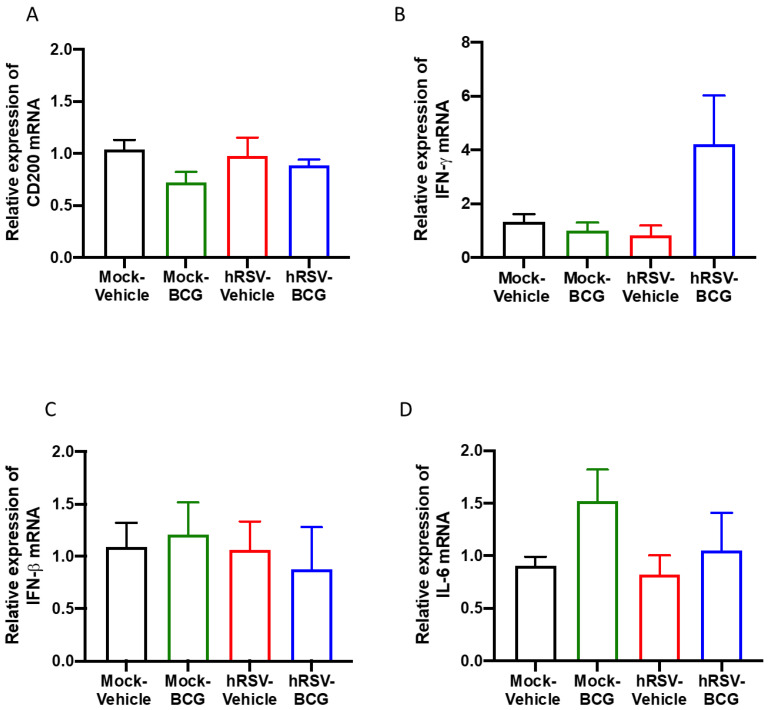
Determination of the relative expression of immunomodulatory molecules and cytokines in infection with Bacillus Calmette-Guerin (BCG) at 10 days post-human respiratory syncytial virus (hRSV)-infection. Quantification by real-time quantitative reverse transcription polymerase chain reaction (RT-qPCR) of OX-2 glycoprotein membrane (*cd200*) at day 10 post-infection (**A**), interferon-gamma (*ifn-γ*) (**B**), interferon beta (*ifn-β*) (**C**), and interleukin (IL)-6 (**D**). Data are shown as means ± SEM of three independent experiments with 3–4 animals per group. One-way ANOVA was performed with a post-hoc Tukey test.

**Figure 5 antioxidants-11-01453-f005:**
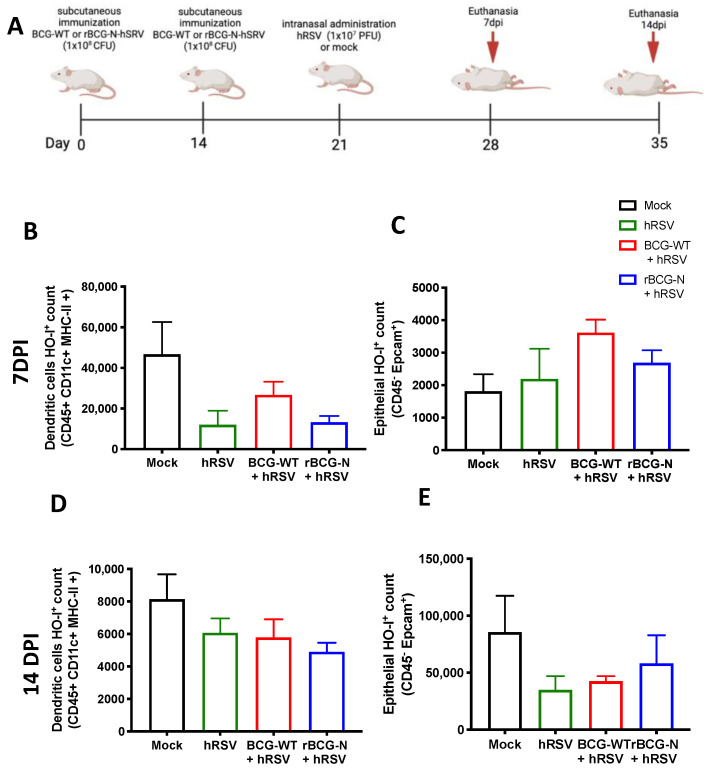
Detection of Heme Oxygenase (HO)-1 in cell populations during an immunization scheme with recombinant BCG expressing the nucleoprotein of human respiratory syncytial virus (rBCG-N-hRSV). Immunization scheme of BALB/c mice (**A**). Analysis by flow cytometry of dendritic cells (DC) (**B**) and epithelial cells (**C**) at 7 days post-challenge, and DCs (**D**) and epithelial cells (**E**) at 14 days post-challenge. Data are shown as means ± SEM of one independent experiment with three animals per group. One-way ANOVA was performed with a post-hoc Tukey test.

**Table 1 antioxidants-11-01453-t001:** Schemes of infection and their times.

	Day of the First Inoculation (Mock or Human Respiratory Syncytial Virus)	Day of the Second Inoculation (Vehicle or *Bacillus* Calmette-Guerin)	Day of the Euthanasia (After the Second Inoculation)
Short scheme	0	10	11
Long scheme	0	21	21

**Table 2 antioxidants-11-01453-t002:** List of primers used for real-time quantitative reverse transcription polymerase chain reaction (RT-qPCR) analysis.

Gene	Forward Primer	Reverse Primer	Gene Accession Code
*n-hRSV*	5′-GCTAGTGTGCAAGCAGAAATC-3′	5′-TGGAGAAGTGAGGAAATTGAGTC-3′	Gene ID: 1489820
Mouse *ho-1*	5′-CCTCTGACGAAGTGACGCC-3′	5′-CAGCCCCACCAAGTTCAAA-3′	Gene ID: 15368
Mouse *nrf2*	5′-TTCTTTCAGCAGCATCCTCTCCAG-3′	5′-ACAGCCTTCAATAGTCCCGTCCAG-3′	Gene ID: 18024
Mouse *ifn-β*	5′-AGCTCCAAGAAAGGACGAACA-3′	5′-GCCCTGTAGGTGAGGTTGAT-3′	Gene ID: 15977
Mouse *il-6*	5′-TAGTCCTTCCTACCC CAATTTCC-3′	5′-TAGTCCTTCCTACCCCAATTTCC-3′	Gene ID: 16193
Mouse *cd200*	5′-CTCTCCACCTACAGCCTGATT-3′	5′-AGAACATCGTAAGGATGCAGTTG-3′	Gene ID: 17470
Mouse *β-actin*	5′-ACCTTCTACAATGAGCTGCG-3′	5′-CTGGATGGCTACGTACATGG-3′	Gene ID: 11461

## Data Availability

The data generated during this study are readily available in the manuscript.

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
