# Peer review of "Increased Heme Oxygenase 1 Expression upon a Primary Exposure to the Respiratory Syncytial Virus and a Secondary Mycobacterium bovis Infection"

_antioxidants, 2022, doi:10.3390/antiox11081453_

Round 1

Reviewer 1 Report

Bibliography

1. Lack of homogeneity in the bibliography citation criteria (there are some citations that have only year and not pages, another have volume and year and pages...). Homogenize please.

2. 53% of the bibliographic citations are more than 10 years old and 13% of them are more than 20 years old. Review, please

Structure

1. Absence of conclusions, in my opinion the conclusions would be what appear from line 470.

2. The acronyms ZN are not defined the first time they are cited (line 196) but are defined in the citation on line 313: please modify.

Author Response

Answer to Reviewer 1

1.- Reviewer 1: Lack of homogeneity in the bibliography citation criteria (there are some citations that have only year and not pages, another have volume and year and pages...). Homogenize please.

Answer: As requested by the Reviewer, the references were homogenized in the reference section (Page 16, Line 596).

2.- Reviewer 1: 53% of the bibliographic citations are more than 10 years old and 13% of them are more than 20 years old. Review, please.

Answer: As requested by the Reviewer, the references were updated or complemented with updated references throughout the manuscript.

3.- Reviewer 1: Absence of conclusions, in my opinion the conclusions would be what appear from line 470.

Answer: As requested by the Reviewer, a section for conclusions was included (Page: 15, Lines: 548-555).

4.- Reviewer 1: The acronyms ZN are not defined the first time they are cited (line 196) but are defined in the citation on line 313: please modify.

Answer: As requested by the Reviewer, the acronym ZN has been defined the first time cited (Page: 5, Line: 216), and the following mentions of ZN were corrected as well.

We would like to thank the Reviewers and the Editors for their time and effort in handling this manuscript and hope that the current revised manuscript is acceptable for publication in Antioxidants.

Reviewer 2 Report

In this article, “Canedo-Marroquin et al.” evaluated, in a mouse model, whether a primary infection by the human respiratory syncytial virus (hRSV), once resolved, dampens the host immune response to a secondary with an attenuated strain of Mycobacterium bovis, BCG. They showed that the previous infection with hRSV has a clear impact, notably through an increased expression of heme oxygenase-1 (HO-1). Even if the subject is interesting (increasing our understanding of the mechanisms that can be induced in co-infections situations) the article is too weak at this stage to be accepted. Indeed, in this mostly descriptive study, the objectives and the methodology are currently not really clear and the article needs a deep English revision. Please find major remarks/corrections below.

Strengths: Interesting subject (co- and super-infections), large amount of work, usually quite well-presented.

Weaknesses: The taking home message of this mostly descriptive study is not crystal clear and the rationale of the link with HO-1 is not always easy to get. The methodology is not very well-presented and there is sometimes a kind of confusion between the BCG infections and the recombinant BCG immunization. Some figures or methodology schemes could help. Then, English must be improved… Some sentences are very long (7 lines) and sometimes really weird, uneasy to understand. A link with “trained immunity” or epigenetic immune modulation could be added.  

Major

-Please revise English and remove typos.

-L32-33: The rationale of the immunization procedure is not clear and must be better explained at the beginning of the article and later.

-L146-149: Some protocols are not clearly presented. Some schemes and or figures would definitely help. Table 1 does not really help and should be improved.

Moderate

-L54: Some references such as doi: 10.1128/CMR.00111-17, doi: 10.1186/s13567-020-00807-8, and doi: 10.1038/s41590-019-0568-x must be added to enrich the introduction and the discussion with co-infection molecular consequences.

-L81: “an enzyme that produces the catabolism?”

-L83-85: This sentence is not clear. Please modify.

-L88?

-L89-92: the justification is not really convincing. Please modify.

-L101-106: This section is not clear and must be rephrased.

-L145: The rationale is not clear and could be briefly introduced here too.

-Table 1: What is the rationale for 11 and 21 days? Please explain in the text.

-L160-174: Authors could provide the working dilutions for the antibodies.

-One single reference gene has been used for RT-qPCR assay. This is not ideal (see MIQE guidelines for qPCR and DOI: 10.1186/gb-2002-3-7-research0034). Is the reference gene stably expressed? Please comment and add a reference in the text.

-Delta delta CT method (it must be referenced in the text: DOI: 10.1006/meth.2001.1262) assumes 100% efficiency of qPCR assays. Was it the case? Please add a sentence regarding that point in the text.

-Statistical analyses (a section should also be added at the end of the material and methods section): Parametric tests need a normal distribution of the data (it can be tested with a Shapiro-Wilk test for instance). Was it the case. It must be explained in the stat section and/or in the legends.

-Authors should provide a table for the primer sequences.

-L280: Not clear, please reformulate.

-L282-283: For the significant differences, please provide the p-values.

-L296: Please provide the numbers of days for the long scheme here.

-L361: The rationale for the immunization is not clear or missing.

-L377: We cannot really talk of a vaccine here.

-In figure 5A: Not easy to make the difference between immunization and challenge. Indeed, BCG is sometimes use to immunize, sometimes to infect… It is quite difficult to follow in the study.

-L400-420: Too speculative…

-L424-425: Except if AM are destroyed, by the infection for instance.

-L424: Please add a reference.

-L446: The last what? Please modify.

Minor

-L2-3: Please avoid abbreviations in title (especially M. which can be Mycobacterium as well as Mycoplasma).

-L22-23, L70: Mycobacterium should be in italic.

-Please remove the “s” in years.

-L50 and elsewhere, the reference numbers should be between the same brackets.

-L125: Please replace ml by mL everywhere.

-Please make sure all the abbreviations are defined, see CFU (L140), PFA (L166), ACK (L171), ZN (L196), dpi, APC (L325)

-L232: Installation?

-Beginning of L244: Please add a link with the figure.

-In figure 1: Please add the number of days for hRSV in panel B.

-L268: Can we really talk about reactivation here?

-L271: Please replace “a” by “any”.

-Figure 2, panel A, too small.

-Figure 2, panel 2D: Some arrows to indicate what we need to see must be added. Please also add in the pictures size bars (also figure 3).

-L305: Something wrong?

-L342-343: This sentence is too vague…

-L368-369: The lowest-non significant level? If not significant, it is only a trend and only a trend should be mentioned. P-value?

-L400: Please add a reference.

-L400-406: Very very long sentence, please split and simplify.

-L406-407: Not clear.

Author Response

Answer to Reviewer 2

1.- Reviewer 2: In this article, “Canedo-Marroquin et al.” evaluated, in a mouse model, whether a primary infection by the human respiratory syncytial virus (hRSV), once resolved, dampens the host immune response to a secondary with an attenuated strain of Mycobacterium bovis, BCG. They showed that the previous infection with hRSV has a clear impact, notably through an increased expression of heme oxygenase-1 (HO-1). Even if the subject is interesting (increasing our understanding of the mechanisms that can be induced in co-infections situations) the article is too weak at this stage to be accepted. Indeed, in this mostly descriptive study, the objectives and the methodology are currently not really clear and the article needs a deep English revision. Please find major remarks/corrections below.

Interesting subject (co- and super-infections), large amount of work, usually quite well-presented.

Answer: We thank the Reviewer for critically revising our manuscript.

2.- Reviewer 2: Please revise English and remove typos.

Answer: As requested by the Reviewer, the grammar and the typos were corrected throughout the manuscript.

3.- Reviewer 2: L32-33: The rationale of the immunization procedure is not clear and must be better explained at the beginning of the article and later.

Answer: As requested by the Reviewer, the rationale for the immunization procedure was included throughout the manuscript.

4.- Reviewer 2: L146-149: Some protocols are not clearly presented. Some schemes and or figures would definitely help. Table 1 does not really help and should be improved.

Answer: As requested by the Reviewer, the protocol was clarified in the main text (Page: 4, Lines: 142-144), table, and figure.

5.- Reviewer 2: L54: Some references such as doi: 10.1128/CMR.00111-17, doi: 10.1186/s13567-020-00807-8, and doi: 10.1038/s41590-019-0568-x must be added to enrich the introduction and the discussion with co-infection molecular consequences.

Answer: As requested by the Reviewer, we included the references in the introduction section and were further discussed in the discussion section (Page: 13 and 14, Lines: 435-437 and 497-498, respectively).

6.- Reviewer 2: L81: “an enzyme that produces the catabolism?”

Answer: As requested by the Reviewer, the word catabolism was corrected (Page: 3, Line: 80).

7.- Reviewer 2: L83-85: This sentence is not clear. Please modify.

Answer: As requested by the Reviewer, the sentence was rewritten for clarification (Page: 3, Lines: 82-84).

8.- Reviewer 2: L88?

Answer: As requested by the Reviewer, the sentence was rewritten to improve clarity (Page: 3, Lines: 87-89).

9.- Reviewer 2: L89-92: the justification is not really convincing. Please modify.

Answer: As requested by the Reviewer, the sentence was modified (Page: 3, Lines: 89-94).

10.- Reviewer 2: L101-106: This section is not clear and must be rephrased.

Answer: As requested by the Reviewer, the section was rewritten to improve clarity (Page: 3, Lines: 97-107).

11.- Reviewer 2: L145: The rationale is not clear and could be briefly introduced here too.

Answer: As requested by the Reviewer, the rationale was clarified (Page: 4, Lines: 142-144).

12.- Reviewer 2: Table 1: What is the rationale for 11 and 21 days? Please explain in the text.

Answer: As requested by the Reviewer, the rationale for 11 and 21 days was included in the main text (Page: 4, Lines: 151-164).

13.- Reviewer 2: L160-174: Authors could provide the working dilutions for the antibodies.

Answer: As requested by the Reviewer, dilutions for the antibodies for flow cytometry experiments were included (Page: 5, Lines: 194-196).

14.- Reviewer 2: One single reference gene has been used for RT-qPCR assay. This is not ideal (see MIQE guidelines for qPCR and DOI: 10.1186/gb-2002-3-7-research0034). Is the reference gene stably expressed? Please comment and add a reference in the text.

Answer: As requested by the Reviewer, a comment and reference were included to justify using a single reference gene (Page: 6, Lines: 234-236).

15.- Reviewer 2: Delta delta CT method (it must be referenced in the text: DOI: 10.1006/meth.2001.1262) assumes 100% efficiency of qPCR assays. Was it the case? Please add a sentence regarding that point in the text.

Answer: As requested by the Reviewer, we included the reference and the efficacy achieved during the RT-qPCR assays (Page: 6, Line: 232).

16.- Reviewer 2: Statistical analyses (a section should also be added at the end of the material and methods section): Parametric tests need a normal distribution of the data (it can be tested with a Shapiro-Wilk test for instance). Was it the case. It must be explained in the stat section and/or in the legends.

Answer: As requested by the Reviewer, we have included the information in the Methods section (Page: 6, Lines: 242-249).

17.- Reviewer 2: Authors should provide a table for the primer sequences.

Answer: As requested by the Reviewer, a list for the primer sequences was included (Page: 6, Line: 237).

18.- Reviewer 2: L280: Not clear, please reformulate.

Answer: As requested by the Reviewer, the sentence was reformulated (Page: 8, Lines: 311-313).

19.- Reviewer 2: L282-283: For the significant differences, please provide the p-values.

Answer: As requested by the Reviewer, we have included the p-value (Page: 8, Line: 315).

20.- Reviewer 2: L296: Please provide the numbers of days for the long scheme here.

Answer: As requested by the Reviewer, the numbers of days for the long experimental scheme were provided (Page: 8, Lines: 327-328).

21.- Reviewer 2: L361: The rationale for the immunization is not clear or missing.

Answer: As requested by the Reviewer, the rationale for the immunization was clarified (Page: 12, Lines: 402-405).

22.- Reviewer 2: L377: We cannot really talk of a vaccine here.

Answer: As requested by the Reviewer, the word vaccination was replaced by immunization (Page: 12, Line: 421).

23.- Reviewer 2: In figure 5A: Not easy to make the difference between immunization and challenge. Indeed, BCG is sometimes use to immunize, sometimes to infect… It is quite difficult to follow in the study.

Answer: As requested by the Reviewer, we have incorporated background to clarify the use of BCG as a vaccine for the trial shown in Figure 5 (Page: 12, Lines: 402-405).

24.- Reviewer 2: L400-420: Too speculative…

Answer: As requested by the Reviewer, the speculative part was modified (Page: 14, Lines: 473-486).

25.- Reviewer 2: L424-425: Except if AM are destroyed, by the infection for instance.

Answer: As requested by the Reviewer, the exception regarding the permanence of alveolar macrophages in the lungs was included (Page: 14, Lines: 495-497).

26.- Reviewer 2: L424: Please add a reference.

Answer: As requested by the Reviewer, a reference was included referring to the permanence of alveolar macrophages in the lungs (Page: 14, Line: 495).

27.- Reviewer 2: L446: The last what? Please modify.

Answer: As requested by the Reviewer, the sentence was modified to improve clarity (Page: 15, Line: 518).

28.- Reviewer 2: L2-3: Please avoid abbreviations in title (especially M. which can be Mycobacterium as well as Mycoplasma).

Answer: As requested by the Reviewer, the abbreviations in the title were replaced by the complete name (Page: 1, Lines: 2-3).

29.- Reviewer 2: L22-23, L70: Mycobacterium should be in italic.

Answer: As requested by the Reviewer, Mycobacterium was rewritten in italic (Page: 1, Line: 26).

30.- Reviewer 2: Please remove the “s” in years.

Answer: As requested by the Reviewer, we removed the “s” in years (Page: 2, Line: 51).

31.- Reviewer 2: L50 and elsewhere, the reference numbers should be between the same brackets.

Answer: As requested by the Reviewer, the reference numbers are now between brackets throughout the manuscript.

32.- Reviewer 2: L125: Please replace ml by mL everywhere.

Answer: As requested by the Reviewer, “ml” was replaced by “mL” (Page 3, Line 123).

33.- Reviewer 2: Please make sure all the abbreviations are defined, see CFU (L140), PFA (L166), ACK (L171), ZN (L196), dpi, APC (L325)

Answer: As requested by the Reviewer, all abbreviations were defined the first time mentioned throughout the manuscript.

34.- Reviewer 2: L232: Installation?

Answer: As requested by the Reviewer, the word was corrected to instillation (Page: 6, Line: 255).

35.- Reviewer 2: Beginning of L244: Please add a link with the figure.

Answer: As requested by the Reviewer, a link with the figure was included (Page 7, Line 268-269).

36.- Reviewer 2: In figure 1: Please add the number of days for hRSV in panel B.

Answer: As requested by the Reviewer, we have included the information in the figure.

37.- Reviewer 2: L268: Can we really talk about reactivation here?

Answer: As requested by the Reviewer, the word reactivation was replaced by hRSV persistence (Page: 8, Lines: 296-298).

38.- Reviewer 2: L271: Please replace “a” by “any”.

Answer: As requested by the Reviewer, “a” was replaced for “any” (Page: 8, Line: 301).

39.- Reviewer 2: Figure 2, panel A, too small.

Answer: As requested by the Reviewer, we have modified the size and resolution of the figure.

40.- Reviewer 2: Figure 2, panel 2D: Some arrows to indicate what we need to see must be added. Please also add in the pictures size bars (also figure 3).

Answer: As requested by the Reviewer, we have incorporated these suggestions in Figure 2D.

41.- Reviewer 2: L305: Something wrong?

Answer: As requested by the Reviewer, the sentence was modified to improve clarity (Page 9, Lines: 336-337).

42.- Reviewer 2: L342-343: This sentence is too vague…

Answer: As requested by the Reviewer, the sentence was modified to state the important role of CD200 in tissue immunity (Page 11, Lines: 380-382).

43.- Reviewer 2: L368-369: The lowest-non significant level? If not significant, it is only a trend and only a trend should be mentioned. P-value?

Answer: As requested by the Reviewer, we have included p-values (Page: 12, Lines: 412-413).

44.- Reviewer 2: L400: Please add a reference.

Answer: As requested by the Reviewer, references were included (Page: 14, Line: 473).

45.- Reviewer 2: L400-406: Very very long sentence, please split and simplify.

Answer: As requested by the Reviewer, the sentence was split and simplified (Page: 14, Lines: 473-478).

46.- Reviewer 2: L406-407: Not clear.

Answer: As requested by the Reviewer, the sentence was rewritten to improve clarity (Page: 14, Lines: 478-479).

We would like to thank the Reviewers and the Editors for their time and effort in handling this manuscript and hope that the current revised manuscript is acceptable for publication in Antioxidants.

Reviewer 3 Report

With interest, I read the manuscript antioxidants-1752963. It is an interesting paper, written by experienced researchers based on a solid study.

Except for some those referring to some parts of statistics, my comments/suggestions are minor/facultative.

Major comments:

1.       Statistics:

a.       Figure 1. “(…) T-student was performed with Mann-Whitney U test (…)”. That does not make sense, either t-test or Mann-Whitney U test. What was used?

b.       In continuation, usage of Mann-Whitney U test would suggest that the data have distribution different from normal, i.e. median with IQR would be a better, more appropriate option.

c.       By occasion, please, do not use commas for decimals (e.g. p-values; throughout the manuscript, incl. the supplement).

d.       Throughout the manuscript, incl. the supplement. It is better to write “a post-hoc Tukey test” than “a posterior Tukey test”.

e.       Suppl. Figure S2F. How was p-value calculated here?

f.        Suppl. Figure S4. Explain significant stars.

Minor comments/facultative suggestions:

1.       All abbreviations need to be explained upon their first appearance, separately in the abstract, main text, and figure/table legends.

2.       I would try to move some parts of the introduction to the discussion, e.g. drawbacks of BCG-based model and some other details. Since the current version of the introduction is long, at the moment, while reaching the final paragraph with study aims, one almost forgets what it is all about.

3.       Figure 1A, 2B, Suppl. Figure S2B. If means with SEMs are shown, try to use only upper or the lower error bar (they are identical). In addition, please, use different colors for different curves. At the moment, it is sometimes difficult to distinguish between them.

4.       In the case of some respiratory viruses, e.g. HRVs (PMID: 32973742, 34067156), downstream epigenetic mechanism are involved. If you find it theoretically possible for RSV and your results, please, discuss.

5.       Graphical abstract would be an added value.

Author Response

Answer to Reviewer 3

1.- Reviewer 3: With interest, I read the manuscript antioxidants-1752963. It is an interesting paper, written by experienced researchers based on a solid study. Except for some those referring to some parts of statistics, my comments/suggestions are minor/facultative.

Answer: We thank the Reviewer for critically revising our manuscript. 

2.- Reviewer 3: Figure 1. “(…) T-student was performed with Mann-Whitney U test (…)”. That does not make sense, either t-test or Mann-Whitney U test. What was used?

Answer: As requested by the Reviewer, we modified the manuscript to clarify that the statistical test used for Figure 1C-F is a Mann-While test since the data do not show a parametric distribution (Page: 8, Line: 285). 

3.- Reviewer 3:  In continuation, usage of Mann-Whitney U test would suggest that the data have distribution different from normal, i.e. median with IQR would be a better, more appropriate option.

Answer: As requested by the Reviewer, we have modified the manuscript to clarify that the Mann Whitney test was only used in Figure 1 and that graphs 1C-F were presented as median with IQR, for the rest of the figures we have used only parametric analyses (Page: 7, Lines: 284-287).

4.- Reviewer 3: By occasion, please, do not use commas for decimals (e.g. p-values; throughout the manuscript, incl. the supplement).

Answer: As requested by the Reviewer, we have changed the "," to "." to report p-values.

5.- Reviewer 3: Throughout the manuscript, incl. the supplement. It is better to write “a post-hoc Tukey test” than “a posterior Tukey test”.

Answer: As requested by the Reviewer, we have modified the information in the figure legends.

6.- Reviewer 3: Suppl. Figure S2F. How was p-value calculated here?

Answer: As requested by the Reviewer, we have included information on p-value statistical analyses (Page: 16, Line: 571).

7.- Reviewer 3: Suppl. Figure S4. Explain significant stars.

Answer: As requested by the Reviewer, we have added information to explain the significant differences found (Page: 11, Lines: 391).

8.- Reviewer 3: All abbreviations need to be explained upon their first appearance, separately in the abstract, main text, and figure/table legends.

Answer: As requested by the Reviewer, all the abbreviations were explained upon their first appearance, separately in the abstract, main text, and figures/table legends.

9.- Reviewer 3:  I would try to move some parts of the introduction to the discussion, e.g. drawbacks of BCG-based model and some other details. Since the current version of the introduction is long, at the moment, while reaching the final paragraph with study aims, one almost forgets what it is all about.

Answer: As requested by the Reviewer, some parts of the introduction were moved to the discussion section (Pages: 13-14, Lines: 431-458).

10.- Reviewer 3: Figure 1A, 2B, Suppl. Figure S2B. If means with SEMs are shown, try to use only upper or the lower error bar (they are identical). In addition, please, use different colors for different curves. At the moment, it is sometimes difficult to distinguish between them.

Answer: As requested by the Reviewer, we have modified the figures accordingly.

11.- Reviewer 3:  In the case of some respiratory viruses, e.g. HRVs (PMID: 32973742, 34067156), downstream epigenetic mechanism are involved. If you find it theoretically possible for RSV and your results, please, discuss.

Answer: As requested by the Reviewer, a possible downstream epigenetic mechanism was discussed (Page: 15, Lines: 539-546).

12.- Reviewer 3: Graphical abstract would be an added value.

Answer: As requested by the Reviewer, a graphical abstract was included (Page: 2, Line: 47).

We would like to thank the Reviewers and the Editors for their time and effort in handling this manuscript and hope that the current revised manuscript is acceptable for publication in Antioxidants.

Round 2

Reviewer 3 Report

My comments have been addressed well.